# Small-scale layered structures at the inner core boundary

Baolong Zhang [1,2], Sidao Ni [1] ✉, Wenbo Wu [3], Zhichao Shen[3], Wenzhong Wang[4], Daoyuan Sun [4] & Zhongqing Wu [4]

The fine-scale seismic features near the inner core boundary (ICB) provide critical insights into the thermal, chemical, and geodynamical interactions between liquid and solid cores, and may shed light on the evolution mechanism of the Earth's core. Here, we utilize a dataset of pre-critical PKiKP waveforms to constrain the fine structure at the ICB, considering the influence of various factors such as source complexity, structural anomalies in the mantle, and properties at the ICB. Our modeling suggests a sharp ICB beneath Mongolia and most of Northeast Asia, but a locally laminated ICB structure beneath Central Asia, Siberia, and part of Northeast Asia. The complex ICB structure might be explained by either the existence of a kilometer-scale thickness of mushy zone, or the localized coexistence of bcc and hcp iron phase at the ICB. We infer that there may be considerable lateral variations in the dendrites growing process at ICB, probably due to the complicated thermochemical and geodynamical interaction between the outer and inner core.

Earth's core plays a key role in the evolution and habitability of our planet. The solid inner core centered at the Earth is surrounded by the most dynamic geosphere of the Earth, i.e., the liquid outer core. Nucleation and solidification of the inner core are believed to release latent heat and light elements into the outer core, driving the geodynamo to generate and maintain Earth's magnetic field[1], while the detailed crystallization mechanism remains unclear[2–6]. The inner core boundary (ICB), where the crystallization takes place, holds the key to understanding the growth, the thermal and compositional evolution of the inner core, and its interaction with the outer core. As a liquid-solid phase transition boundary, the ICB is assumed to be a sharp and flat interface[7,8]. However, previous seismological studies have revealed complex inner core structures below the ICB, including the hemispherical and strong regional variations in isotropic and anisotropic heterogeneity structures[9–11], suggesting a rather complicated crystallizing and melting process[4].

Over the past two decades, previous studies have reported increasing evidence for rather complicated fine-scale structures at ICB (i.e., a few to tens of km)[11,12]. For example, the small-scale irregular topography of ICB has been reported by fitting the travel time, amplitude, spectra, waveform, and coda observations of body waves incident on the inner core[13–16]. Moreover, a lateral variation of ICB properties (velocity and density contrast) have been revealed by PKiKP/PcP and PKiKP/P ratios, scattered seismic waves[17–21]. Krasnoshchekov et al.[22] reported significant amplitude variations of the PKiKP phase (P wave reflected from ICB) at some seismic stations from the Semipalatinsk array recording explosions and other nuclear tests conducted during the 20th century, which were interpreted to be attributed to the mosaic structure at the ICB with a thin partially liquid layer interspersed with patches containing a sharp transition. Recently, the presence of a thin (several kilometers thick) mushy zone or irregular transition layer on the inner core's surface was proposed to account for the amplitude and waveform anomaly of reflected waves at ICB[23,24]. These seismic observations may be related to the thermal and chemical variations at the ICB. However, the fine structure of the ICB is still poorly understood, due to the limited seismic observations sampling the ICB, precluding the understanding of thermochemical and geodynamical interaction between the outer and inner core. High-

[1]State Key Laboratory of Geodesy and Earth's Dynamics, Innovation Academy for Precision Measurement Science and Technology, Chinese Academy of Sciences, Wuhan 430077, China. [2]CAS Center for Excellence in Deep Earth Science, Guangzhou, China. [3]Department of Geology and Geophysics, Woods Hole Oceanographic Institution, Woods Hole, MA 02543, USA. [4]Laboratory of Seismology and Physics of Earth's Interior, School of Earth and Space Sciences, University of Science and Technology of China, Hefei 230026, China. ✉e-mail: sdni@whigg.ac.cn

frequency pre-critical PKiKP waves are highly sensitive to small-scale heterogeneities, seismic property contrasts, and topography variations on the ICB[13,14,18,22,23].

Here, we present a new dataset of pre-critical PKiKP waveforms (at distance less than 50°) of deep earthquakes recorded at small aperture seismic arrays and regional networks to depict the fine structure characteristics at the ICB beneath local regions that have not been imaged yet. We model the velocity structure at the ICB by fitting the observed PKiKP waveforms and further infer the crystallization mechanism of the inner core.

## Results

### PKiKP observations

Our study area is located beneath central and eastern Asia, which is characterized by a number of dense seismic arrays and relatively high seismicity. Several small aperture dense arrays including the Kurchatov-Cross Array (KURA), Karatau Array (KKA), and Borovoye Array (BVA), have been in continuous operation over decades (Fig. 1a and Supplementary Fig. 1). Besides these small aperture arrays, the XL temporary seismic array was deployed in Central Mongolia from 2012 to 2016. Moreover, there are also permanent seismic networks in this region, including the China National Seismic Network. To mitigate potential contaminations on the PKiKP and PcP caused by the complexity of earthquake rupture, early aftershocks, and lithospheric scattering, we carefully scrutinize all the seismic recordings of moderate events ($5.7 \leq MB \leq 7.0$) deeper than 80 km[23,25]. Because stronger earthquakes have complex source time functions, and small events

might be too weak to excite observable PcP and PKiKP waves. Nuclear explosions are also included because they are highly impulsive volumetric sources, hence favorable for exciting the core seismic phases PKiKP and PcP[22,26,27]. We collected data on over 440 events occurring between January 1994 to December 2021 within the regions spanning 15-55 N° and 65-155 E°, which are recorded by those seismic arrays within epicentral distances of 50°. These earthquakes are mostly distributed across the Hindukush, Myanmar, and the western Pacific subduction zone regions.

After the following screening criteria, we collect more than 400 high-quality traces of waveform with both clear PcP and PKiKP phases in the frequency range of 2–3 Hz from four events (Supplementary Table 1), which mainly sampled the ICB beneath Central Asia, Siberia, and Northeast Asia. Both PKiKP and PcP waves share similar raypaths in the crust and mantle, using PcP as reference phase accounts for the source effect, lithospheric complexity beneath seismic stations, and mantle anomalies along the PKiKP ray path (Fig. 1a). To select robust observations of PKiKP and PcP with high signal-to-noise ratios (SNR), we retain waveforms either with an SNR ≥ 2.0 for single traces or coherent signals on record sections observed at dense arrays. SNR is defined as the ratio of the peak amplitude of PcP or PKiKP to the ambient noise amplitude of the waveform window 5 to 10 s before the PcP or PKiKP arrival. We excluded most of the seismograms from further processing due to the high noise levels significantly hinder the detection of PKiKP signals. For events with high SNRs in their PcP and PKiKP seismograms, we additionally performed manual inspections to assess the quality of valuable observations from small-aperture dense

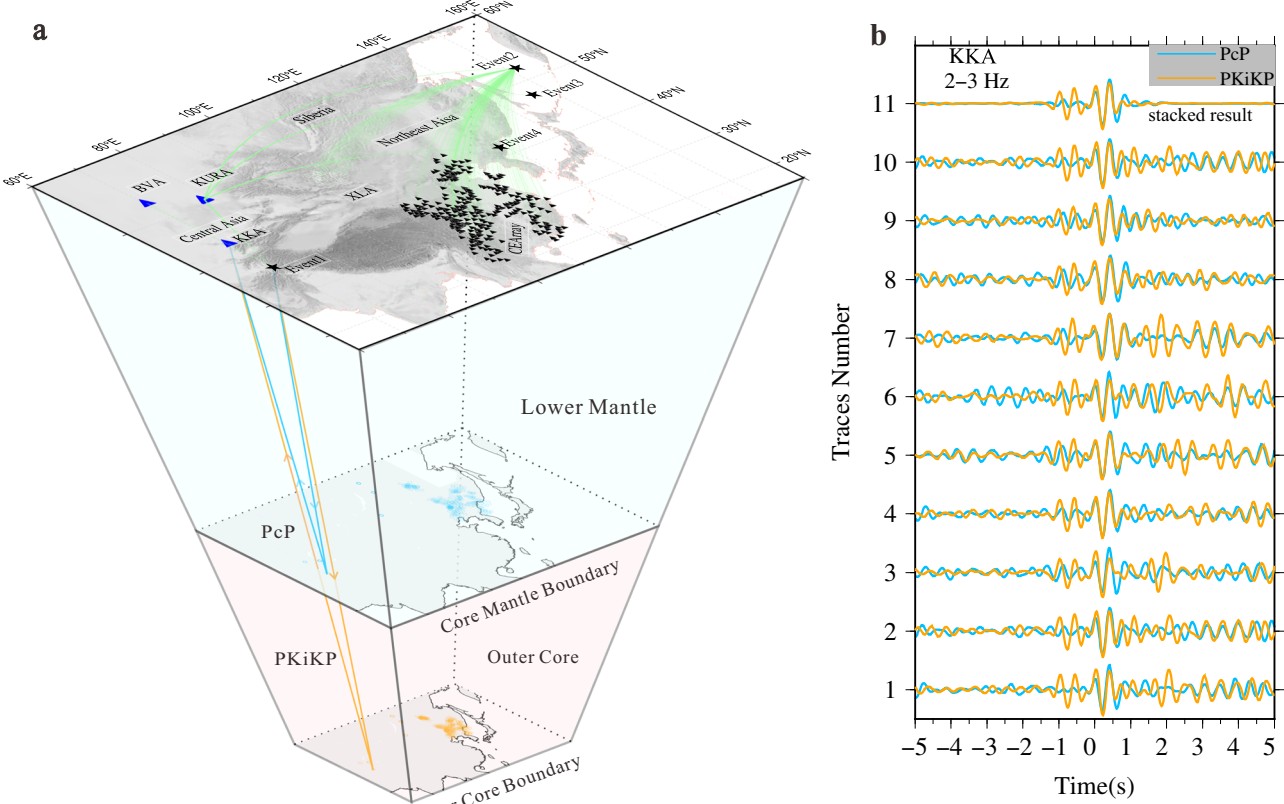

**Fig. 1 | Study region and an example of PKiKP and PcP observations. a** Seismic raypaths (blue for PcP and orange for PKiKP) along with great circle paths (green lines) connecting seismic stations (triangles) and events (stars, the information of events used in this study is shown in Supplementary Table 1). PcP and PKiKP reflection points are indicated with blue and orange circles on the core-mantle boundary (CMB) and inner core boundary (ICB), respectively. The blue triangles are small aperture dense arrays, the detailed array configurations of the Kurchatov-

Cross Array (KURA), Karatau Array (KKA), and Borovoye Array (BVA) are shown in supplementary Fig. 2. **b** Comparison of PKiKP and PcP waveforms at 2.0 to 3.0 Hz at the KKA array for Event 1. Partial velocity seismograms of PcP (blue) and PKiKP (orange) observations are plotted with normalized amplitude, and the 11th trace is the stacked result. The time zero represents the travel time of the PKiKP and PcP phases.

arrays. We found that despite some seismic traces having an SNR below 2, PKiKP and PcP signals were coherent, thus still discernible. Therefore, we retain those visually identifiable PcP and PKiKP signals recorded by small-aperture arrays in Central Asia from these four events in Supplementary Table 1. Furthermore, we determine slowness and backazimuth to verify the reliability of the identifying PcP and PKiKP phases in Supplementary Fig. 2 and Supplementary Fig. 3 (see Methods section).

Furthermore, certain seismic phases with different propagation paths may exhibit travel times close to those of the PKiKP or PcP at some epicentral distances (Supplementary Fig. 4). Although we could suppress those interfering signals via bandpass filtering or array stacking technologies, the contaminated PKiKP waveforms are difficult to be fully recovered. Such contamination could be misinterpreted as the result of small-scale heterogeneity at the ICB. Therefore, seismic observations of the PcP and PKiKP phases were excluded from analysis when in proximity to interfering waves, such as S and ScS. For instance, in the Hindu Kush region, there are several moderate events with depths of around 200 km excited high-quality PKiKP waveforms. But the corresponding reference phase PcP at the dense arrays in Central Asia was severely contaminated by the strong S-coda waves. Good records of PKiKP and PcP phases are simultaneously observed only for Event 1. Along the ray path from Myanmar to Central Asia, where the epicentral distance is approximately 30°, the PKiKP waves were significantly impacted by high-frequency ScS waveforms (Supplementary Fig. 4a).

## Low-velocity layers at the inner core boundary

The PKiKP waveforms at the KKA array from Event 1 exhibit clearly different waveforms compared with PcP (Fig. 1b). An extra signal arrives at ~1 s before the predicted PKiKP major arrival apart, while it is missing in the PcP across the entire array. To further enhance the more coherent PcP and PKiKP signals, we stacked these array data based on the time-frequency domain phase-weighted stack (tf-PWS) method[28]. We computed the spectrograms of the array observations using the S-transform[29] to analyze frequency dependence and energy spectrum of stacking coherent signals. Then, inverse S-transform was performed on those time-frequency domain array observations to obtain the stacked waveforms. At frequencies above 2.0 Hz, the spectrograms of the stacked PKiKP show significant bifurcation for raypaths sampling the ICB beneath Central Asia and Siberia (Fig. 2b, d, f and Supplementary Fig. 5a), whereas the corresponding PcP spectra remain intact (Fig. 2a, c, e), consistent with that at individual stations (Fig. 1b). We replicated this processing for array data at XL array (XLA) from the Event 2 (Fig. 1a) and KURA array from the Event 4, and found similar spectrograms of stacked PcP and PKiKP (Fig. 2g, h and Supplementary Fig. 5b). Overall, the disparity in the shape of PcP and PKiKP waves in spectrograms is primarily observed within the frequency range of 2.0 Hz to 4.0 Hz (Fig. 2 and Supplementary Fig. 5a). However, the significant energy of the PcP and PKiKP at the KURA from Event 4 (nuclear explosion) is below 2.5 Hz (Supplementary Fig. 5b), a bandpass filter of 1.5–3.0 Hz was used for this observations.

The stacked results for all these four events are classified into two groups: one group has highly similar PcP and PKiKP waveforms (Fig. 3b) and the other group shows rather complicated PKiKP (Fig. 3c). The former group would be explained by the simple core-mantle boundary (CMB) and ICB structures with a lack of roughness and high sharpness, or corresponding to a gradual transition zone with a thickness less than 5 km as tested in Supplementary Fig. 6. The other group of complicated PKiKP is reflected at the ICB beneath Central Asia, Siberia, and partial of Northeast Asia (six solid boxes in Fig. 3a). Overall, Event 1 in the west shows significant double-peak PKiKP waveforms with a time separation of ~1.0 s on three small-aperture dense arrays in Central Asia, while the other three events in the east

either feature a simple ICB or have PKiKP waveforms with a few extra cycles at the end in the frequencies above 2 Hz.

To facilitate observing the complexity of PKiKP waveforms, it is desirable for the reference phase PcP waveform to exhibit a relatively simple feature. However, PcP may be affected by anomalous structures near the CMB, leading to a more complex PcP waveform compared to PKiKP. For instance, the PcP waveforms recorded by the Makanchi array at Kazakhstan for Event 4 exhibit significantly greater complexity compared to the corresponding PKiKP waveforms in the frequency band of 1.5–3.0 Hz in Supplementary Fig. 7. Consequently, we have excluded observation data characterized by more intricate PcP waveforms. To explain the distorted PKiKP waveforms, we adopted the direct solution method (DSM) to calculate the short-period PcP and PKiKP synthetic seismograms for the IASP91 model[30,31]. By comparing the PcP synthetic with the stacked PcP observations in Supplementary Fig. 8, and found that the synthetic seismogram can fit the observed data well with a correlation coefficient of above 0.78, indicating that the reference seismic phase PcP waveform can be well explained by the 1D model of IASP91.

As our observations at a maximum epicentral distance are ~43.8°, the separation between PcP and PKiKP sampling point at the 660-km discontinuity is about 1.42°, and therefore raypaths of PcP and PKiKP phase in the crust and upper mantle are very similar. This suggests that they were influenced by similar source complexity and the 3D structure of the crust and upper mantle. The ray paths of PcP and PKiKP deviate substantially in the lower mantle, especially at the bottom of the mantle, and spatial separation between PcP and PKiKP sampling point at the CMB is in the range of ~2.5°–17.2°. Previous studies suggested that there is strong lateral heterogeneity in the D″ layer above the CMB, which might cause PKiKP waveform distortion[32]. However, numerical experiments show that the anomalous CMB structure has a much less effect on PKiKP waveforms than the corresponding PcP waveforms[33], which may cause a more complex PcP waveform compared to PKiKP. The PKiKP seismic phase has a long propagation path in the liquid outer core, but it is widely accepted that the highly dynamic outer core is very homogeneous, thereby exerting little influence on the PKiKP waveforms[34,35]. Instead, the small-scale scatterers at the top of the inner core may excite a long random coda tail behind the PKiKP phase[20,36,37], but it is difficult to form coherent double peaks in the PKiKP waveforms at the seismic arrays. Therefore, PKiKP waveform distortion observed in this study is probably caused by the seismic anomalies near the ICB, and a more detailed analysis will be conducted in the discussion section regarding the effects of source complexity, as well as structural anomalies within the lithospheric, upper mantle, and lower mantle on PKiKP and PcP waveforms.

Previous studies suggested that a laminated ICB structure may cause PKiKP waveform distortions[23,38]. To unravel the complexity of the stacked PKiKP waveforms in Fig. 3c, we performed a series of numerical tests. The following strategies are adopted to improve the calculation efficiency and accuracy of waveform modeling. A grid search based on the reflectivity method is performed to find the optimal parameter combination (see Methods section). However, given that the reflectivity method does not calculate multiple reflected waves, we then calculate the short period (with a dominant frequency of about 4 Hz) PKiKP synthetics with the numerical algorithm DSM to fit the stacked PKiKP, based on those optimal parameter combinations (Supplementary Fig. 9) to find the best fitting. From the optimal model presented in Fig. 4, we found that three regions beneath Central Asia, Siberia, and part of Northeast Asia (Fig. 3a), have an anomalous thin layer with a thickness of about 2.2–5.0 km at the ICB. The velocity and density jumps of these layers are about 20–60% of the IASP91 model ICB seismic property jumps (Supplementary Fig. 9a–d). The cross-correlation coefficient between the synthetics and the observations is larger than 0.84, indicating that a double-layered boundary model can well explain the distortions of PKiKP waveforms sampled in these

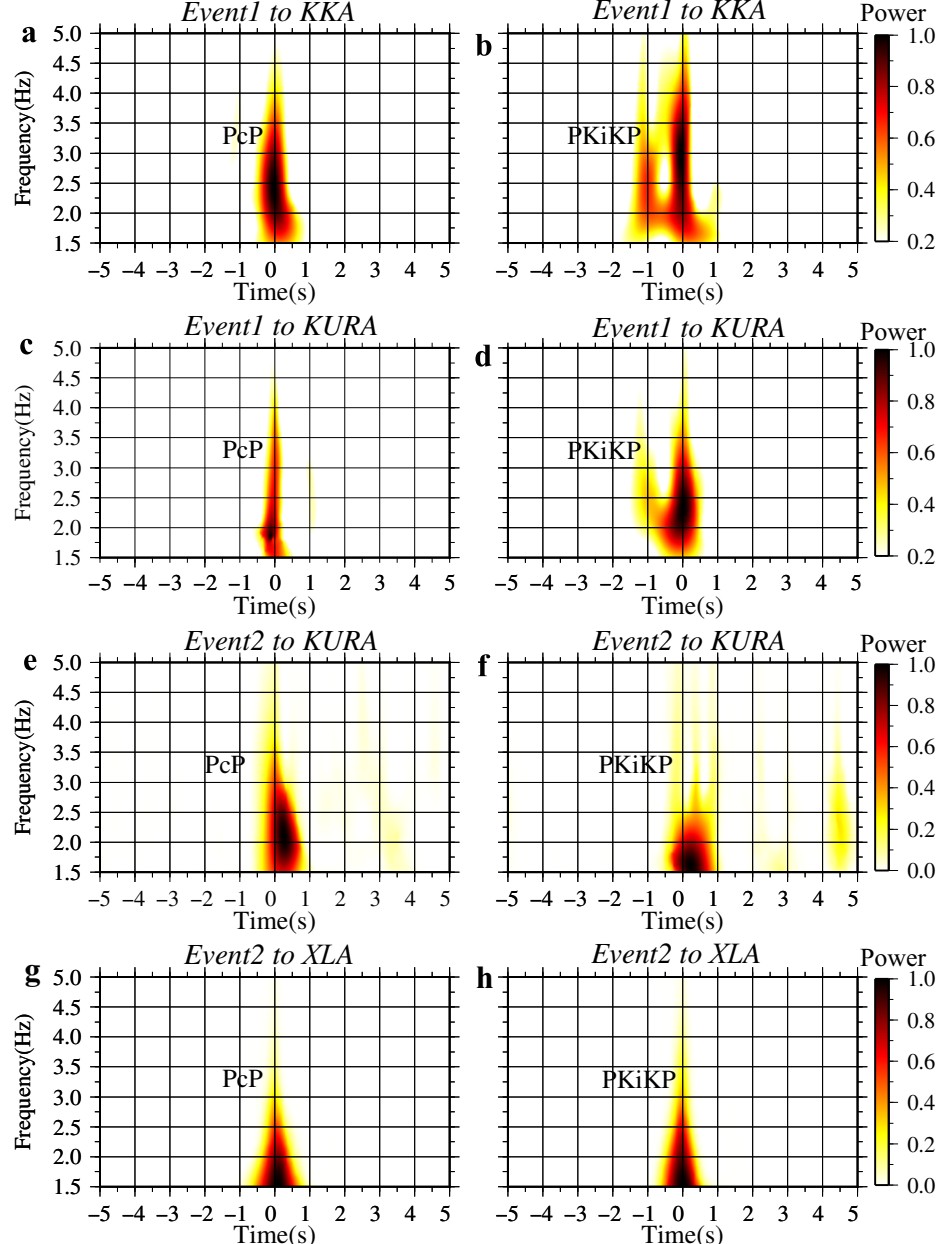

**Fig. 2 | Spectrograms of stack PcP and PKiKP at different seismic arrays.** Shown are (**a**, **b**) the spectrograms of stack PcP and PKiKP at KKA from Event1. (**c**, **d**) are the same as (**a**, **b**), but for the array KURA from Event1. (**e**, **f**) are the spectrum results of PcP and PKiKP respectively for the array KURA from Event2. (**g**, **h**) are the spectrum results for the array XLA from Event2. The time zero is the PcP and PKiKP arrival time. KKA, KURA, and XLA are the Karatau Array, Kurchatov-Cross Array, and XL temporary seismic array, respectively.

regions. However, when a grid search was performed on the IA2 and IA3 regions of Central Asia based on double-layered ICB models, we found that the correlation coefficient was less than 0.8 (Supplementary Fig. 10a), and the synthetics cannot well fit the observed PKiKP waveforms, especially at the part of first weaker wiggle (Supplementary Fig. 10b). We then performed the grid search utilizing triple-layered ICB models. For a top layer is about 2 km thick with velocity and density jumps ranging from -10% to -20%, and a middle layer is ~5 km with jumps from -45% to -55% (referred to as ModelICB3a, in Fig. 4a and Supplementary Fig. 9e, f), the synthetic could fit well with the observations at IA2 and IA3 regions, exhibiting the cross-correlation coefficients of 0.93 and 0.88 (Fig. 4b), respectively. However, with the thickness of ~4.5 km and ~2.2 km for the first and second layers (referred to as ModelICB3b), the cross-correlation coefficient was still near 0.90 (Supplementary Fig. 9f). Upon comparison of the

stacked PKiKP waveform with the synthetic calculated using ModelICB3b, we found that although the phase fitting was satisfactory, the waveform amplitude fitting was relatively poor (Supplementary Fig. 11). Thus, ModelICB3a could explain the stacked PKiKP data sampling IA2 and IA3 regions better than the ModelICB3b.

The ICB model shown in Fig. 4a is constructed based on PKiKP modelings in the frequency band of 2–3 Hz. We also obtained high-quality PKiKP observations at higher frequencies (3–4 Hz) in both IN2 and IN4 sampling areas. By comparing synthetics and observation, we found that the triple-layered boundary model in the IN2 region and the double-layered boundary model in the IN4 region can also explain their corresponding observations in the 3–4 Hz filter band (Supplementary Fig. 12) and two non-overlapping frequency bands, implying the reliability of the modeling results. Overall, our modeling indicates a laminated ICB beneath Central Asia, Siberia,

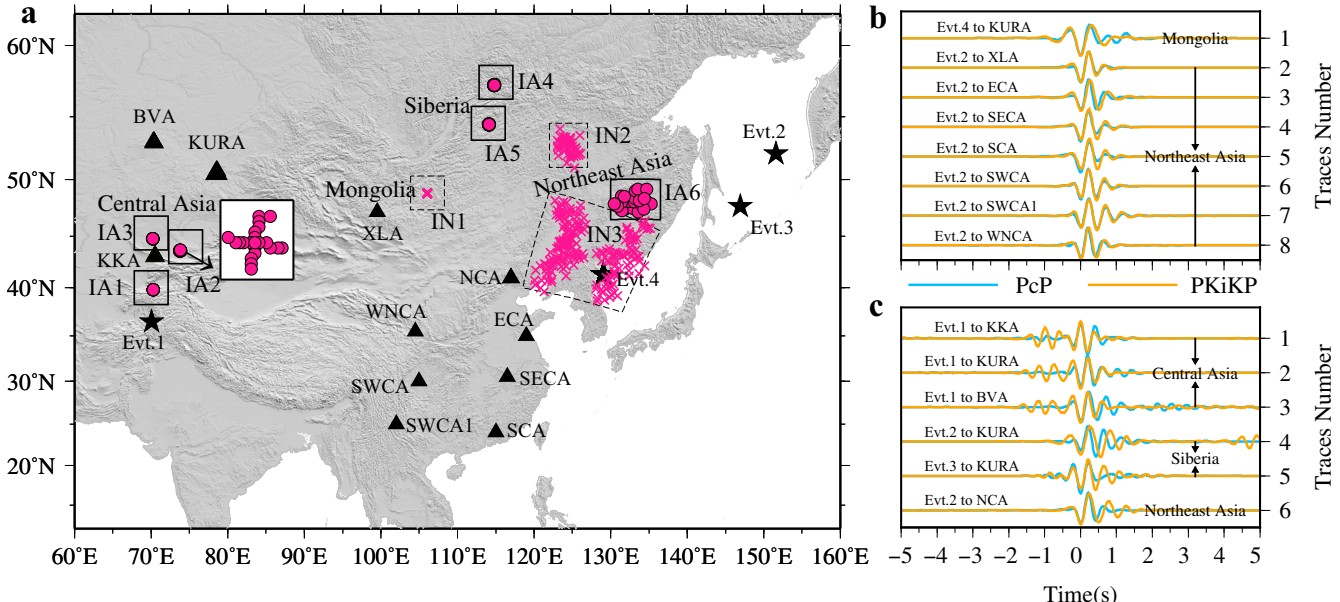

**Fig. 3 | Comparisons of stacked PKiKP and PcP waveforms for different arrays from 4 events. a** Geographical distribution of the inner core boundary (ICB) reflections. Evt. is the abbreviation of Event. Red crosses denote the reflected points where the stacked PKiKP waveforms are similar to the corresponding PcP (labeled as IN1 to IN3 regions with dash boxes, where IN denotes ICB normal region). Whereas red circles denote the reflected points where the stacked PKiKP and the corresponding PcP waveforms show significant differences (labeled as IA1 to IA6 regions with black boxes, where IA denotes ICB anomalous region). Triangles and stars respectively denote seismic arrays and Events. **b** Comparisons of stacked PKiKP and PcP waveforms in the dash boxes region. **c** Comparisons of stacked PKiKP and PcP waveforms in the black boxes regions. Orange and blue traces in (**b, c**) are the stacked PKiKP and PcP waveforms, respectively. Each trace in (**b, c**) represents the stacked waveforms of a dense array from one event.

and partial of Northeast Asia, and a sharp ICB beneath Mongolia, most of Northeast Asia.

## Discussion

The complexity of seismic sources might hinder the reliable modeling of PcP and PKiKP. Thus the source time function (STF) is important for analyzing waveform anomalies of PcP and PKiKP. According to Supplementary Fig. 13, Event 1 displays a simple STF lasting around 1 s. Despite Event 2 having a magnitude of M6.7, its STF duration was only about 2 s, indicating a relatively simple source time function, possibly due to it being a deep earthquake with supershear rupture characteristics. However, the source time functions (STFs) of Events 3–4 exhibit obvious complexity compared to their corresponding observed PcP and PKiKP waves. Due to the shallow source depth of the nuclear explosion, the surface reflected wave pP will affect the direct P wave, resulting in a negative pulse in the STF of Event 4. And it is possible that only a portion of the STF of Event 3 is present in PcP and PKiKP phases. That might make the PKiKP waveforms more sensitive to the take-off angle and azimuth, giving more possibilities to explain the observed differences and posing difficulties in interpreting the absolute and relative amplitudes.

Earth's near-surface layers including the lithosphere and upper mantle have a strong small-scale heterogeneity[39], which may affect the amplitude and waveforms of PKiKP and PcP. For instance, Tkalčić et al.[17,32] calculated an abundant record of PcP and PKiKP travel times and amplitudes from a single earthquake and a nuclear explosion, quantitatively analyzing the influence of small-scale heterogeneities near the surface regions on the amplitude ratio of PKiKP/PcP. They found some complexities and differences between the PKiKP and PcP in individual seismograms that could originate from the upper mantle and lithosphere. Although there is no direct overlap of ICB reflection points between their and our studies, our observations of PcP and PKiKP might still be influenced by similar shallow structures due to the proximity of some of our stations in Central Asia (Supplementary Fig. 14a). Previous seismological studies demonstrated that the array stacking technique could effectively enhance the SNR of core phases such as PcP and PKiKP while suppressing scattering waves from strong small-scale heterogeneities near the surface[22,23,40]. There has been an increasing accumulation of seismic array observational data in recent years, which have provided a solid foundation for the application of station stacking techniques to image the fine-scale structure of the inner core[11]. Therefore, to reduce the effects of near-surface structure, we have stacked these array PcP and PKiKP data based on the time-frequency domain phase-weighted stack (tf-PWS) method in our study.

The amplitude ratios of PKiKP and PcP in the time domain are widely used to constrain ICB properties[16–18,21]. However, significant variations in the amplitude ratios of PKiKP/PcP are observed across different stations, including within the small-aperture dense arrays (Supplementary Fig. 14b). These variations may be caused by multi-scale heterogeneities along the propagation paths of PKiKP and PcP, as well as disparities in instrument performance and installation conditions. Theoretically, in regions where the ICB exhibits small-scale layered structures, the observed PKiKP/PcP ratios should be smaller than the theoretical predictions from the IASP91 model. From Supplementary Fig. 14b, it is indeed evident that the amplitude ratios of PKiKP/PcP in the Central Asia region (IA1), Siberia (IA4 and IA5), and partial of Northeast Asia (IA6) are significantly smaller than the theoretical values of the IASP91 model. In regions such as Mongolia and most of Northeast Asia, where the ICB is considered to be normal, the amplitude ratio of PKiKP/PcP is comparable to the theoretical values. However, in the anomalous ICB regions such as IA2 and IA3 beneath Central Asia, their PKiKP/PcP amplitude ratios are larger than the theoretical value due to the SNR of PcP phases being relatively low in this region.

The thermal control from the lateral heterogeneous lower mantle may drastically affect the outer flow, which in turn produces textural fine-scale heterogeneity on the inner core solidification front[4,41,42]. Probably, dendrites may grow more rapidly in localized regions at the

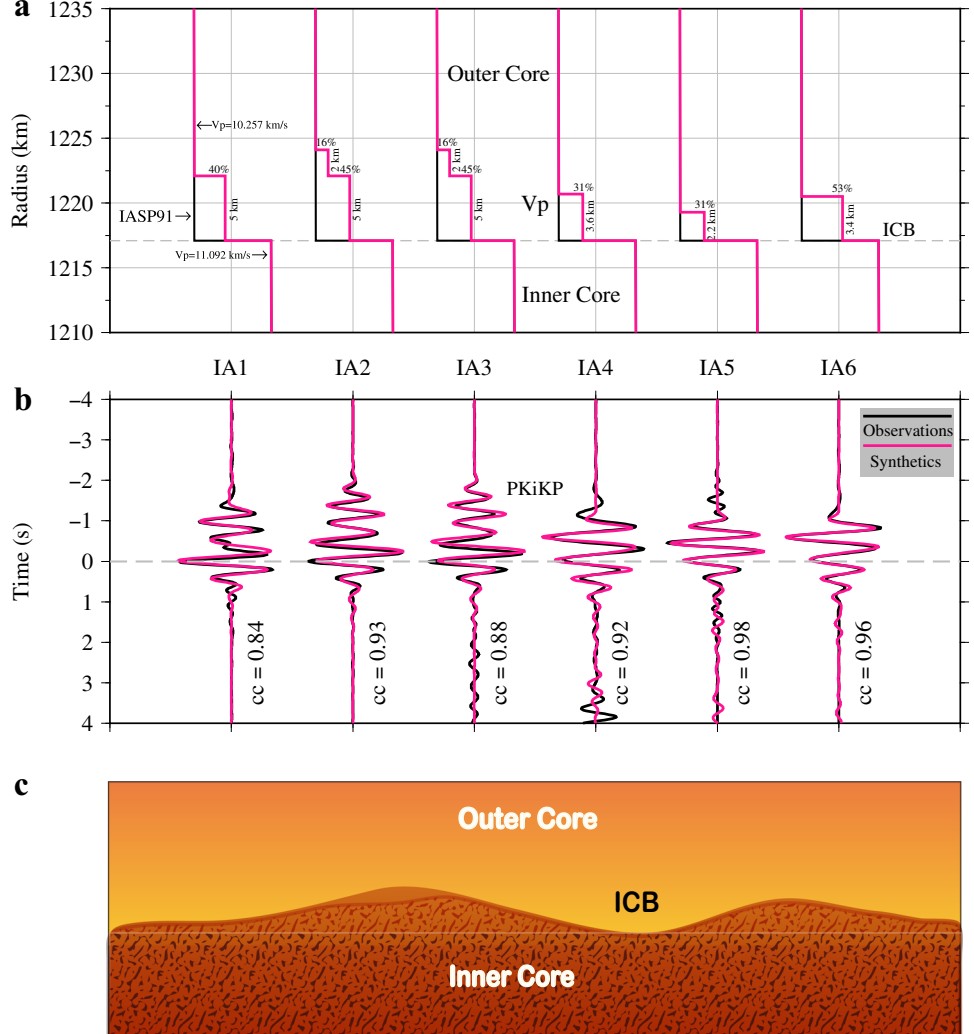

**Fig. 4 | Waveform modeling for observed PKiKP waveforms sampling in anomalous inner core boundary (ICB) regions. a** The best-fitting models (red lines) were adopted in (**b**), the black lines are the iasp91 model, gray dash line is the ICB. **b** Synthetic seismograms of best-fitting models (red) and observed PKiKP waveforms (black) sampling in IA1, IA2, IA3, IA4, IA5, and IA6 regions in Fig. 3a (IA denotes ICB anomalous region), cc is the correlation coefficient. **c** Sketch of the ICB beneath the study region.

inner core surface[14] or with small-scale dynamic forces deforming the ICB[13], resulting in an ICB with a rough boundary[12–15,43]. Seismologists have demonstrated that the existence of small-scale topography of ICB could affect the amplitude of pre-critical PKiKP[13,14]. We use a 2D finite difference (FD) method[44] to simulate the PKiKP wave reflected at a rough ICB. A simple sinusoidal bump and dip ICB topography anomalies (Supplementary Fig. 15a, b) are set up at a distance range of 5.3–7.1°, with a height or depth of 5 km. The PKiKP synthetics at the distances of 10.6–14.2° could sample the ICB anomalies based on raytracing experiments. At the frequency band of 2–3 Hz, we found that the double-peak PKiKP are observed in synthetic waveforms at a larger distance range with both bump and dip ICB topography models (Supplementary Fig. 15a, b), due to the multi-pathed PKiKP and a larger Fresnel zone radius (r ~100 km with a central frequency of 2.5 Hz) for PKiKP at the ICB. The effect of ICB topography on the PKiKP waveform is complex, and it is difficult to quantitatively determine the location and size of topography anomaly by waveform modeling. The distortion observed in the stacked PKiKP in Fig. 3c may result from the combined effect of anomalies at ICB and lateral variation within the uppermost inner core (Fig. 4c).

As a hypothesis of the inner core growth process, the inner core solidification may be dendritic iron crystal growth from the ICB

surface, and may eventually develop the mushy layer and small-scale topography of the ICB[2,14]. Previous core dynamic studies suggest that liquid close to ICB is supercooled in a localized mushy zone where solid dendritic structure coexists with a solute-rich liquid[2,3,45–48]. As a reactive porous medium of the mixed-phase mushy zone, its seismic velocity and density may be intermediate between those of the outer and inner core. The thermodynamic thickness of the mushy layer was predicted to exceed 100 km depending on the phase diagram of the core mixture early[45]. However, it was argued that such a thick mushy zone would collapse under its own weight, and eventually form a thin layer with a thickness of no more than 1 km at the surface of the inner core[3,46], making the ICB appear seismically sharp. Recently, geodynamicists presented that a thick enough (langer than 6 km) mushy layer would be promoted by an inner core viscosity larger than $10^{22}$ Pa.s[49]. By measuring the velocity and attenuation characteristics of the top inner core, Cao and Romanowicz[50] suggested the existence of a mushy zone at the surface of ICB. Seismologists provided evidence for a localized mushy layer at the ICB with a varying thickness of 4–10 km from observations of pre-critical PKiKP and antipodal PKIIKP waves[23,24]. Furthermore, ICB anomalies beneath Central Asia closely match the locations of some sampling points documented in the previous study (Supplementary Fig. 14a)[22], indicating the reliability of the result.

Therefore, the small-scale layered structures at ICB beneath Central Asia, Siberia, and partial of Northeast Asia could be interpreted as the mushy zones.

In addition, the low-velocity layered structures above the ICB may be related to the nucleation of iron during the inner core crystallization. A recent study has investigated the inner core crystallization process using a persistent embryo method and molecular dynamics simulations[6]. They found that the metastable, body-centered, cubic (bcc) iron has a much higher nucleation rate than the hcp iron under inner-core conditions, suggesting that the bcc nucleation may be the starting step of inner core formation, rather than direct nucleation of the hcp phase. As the hcp iron has lower free energy than the bcc iron[51], the bcc structure would finally transform to hcp if time is long enough to reach equilibrium. This two-step nucleation scenario for inner core formation suggests that there may be a metastable phase transition from bcc to hcp iron at the ICB. Based on the elastic properties of bcc and hcp iron calculated using ab initio molecular dynamics under Earth's inner core conditions. The P-wave and S-wave velocities of bcc-Fe are 6.8% and 14.5% lower than hcp-Fe at inner core conditions[52]. This indicates that there may be two velocity jumps above the ICB as a result of the melt-bcc and bcc-hcp phase transitions. Such a scenario can explain the kilometer-scale thickness of lower velocity layer structures at the top of the inner core beneath Central Asia, Siberia, and the partial of Northeast Asia. The presence of metastable bcc iron nucleation may be related to the thermal and chemical anomalies at the regional ICB. However, the mechanism for the bcc nucleation formation has only been investigated in the pure iron system, and it is unknown how light elements affect the formation of nucleation. Also, the partition of light elements between melt and solid phases may also significantly affect the phase boundaries and the velocity and density jump across them. Moreover. the heat flow from the inner core could potentially be subject to local influences stemming from the laminated ICB structure. Future mineral physics studies on these aspects will benefit our understanding of the layered structures observed at the top of the ICB and further interpretation of the inner core formation mechanism.

In conclusion, we use array-processing methods for a new dataset of pre-critical PKiKP waveforms (at distance less than 50°) of deep earthquakes recorded at small aperture dense arrays and regional networks. We found PKiKP waveforms are more complicated than the PcP in both spectrograms and stacked seismograms, which sampled the ICB beneath Central Asia, Siberia, and Northeast Asia. Whereas those PKiKP waveforms sampling the ICB beneath Mongolia, most of Northeast Asia are similar to the corresponding PcP waveforms. After analyzing the influence of factors such as the source complexity, mantle and ICB structural anomalies on the PKiKP waveforms, we propose that the PKiKP waveform distortions in this study are caused by the seismic anomalies near the ICB. Our modeling suggests a laminated ICB beneath Central Asia, Siberia, and partial of Northeast Asia, and a sharp ICB beneath Mongolia, most of Northeast Asia, which appears to feature kilometer-scale (~2.0–~7.0 km in thickness) of low-velocity layered structures at the top of the inner core. The fine-scale laminated ICB structure might be explained by either the existence of mushy zone, or the localized coexistence of the bcc and hcp iron phase at the ICB. We infer that there may be considerable lateral variations in the dendrites growing process at ICB, probably due to the complicated thermochemical and geodynamical interaction between the outer and inner core.

## Methods
### Measuring PKiKP and PcP slowness
Slowness and backazimuth are essential in verifying the reliability of the identifying PcP and PKiKP phases. Therefore, we computed the vespagrams for these seismic arrays using a nonlinear stacking method (Nth-root process) and the slowness of the observed signals that are similar to the theoretical predictions of PcP and PKiKP (Supplementary Fig. 2). Furthermore, we determine the slowness and backazimuth of the observed signals at different seismic arrays with the frequency-wavenumber analysis technique[40]. From Supplementary Fig. 2 and Supplementary Fig. 3, We found that the slowness measurements obtained from both methods are very consistent. Additionally, the back-azimuth measurements of PcP and PKiKP signals are also close to their great circle paths.

### PKiKP-PcP travel time residuals measurements
The PKiKP-PcP differential traveltime residual is defined as:

$$\Delta T^{\text{PKiKP}-\text{PcP}}_{\text{obs}-\text{iasp91}} = \left( T^{\text{PKiKP}}_{\text{obs}} - T^{\text{PcP}}_{\text{obs}} \right) - \left( T^{\text{PKiKP}}_{\text{iasp91}} - T^{\text{PcP}}_{\text{iasp91}} \right) \quad (1)$$

Where the $T^{\text{PKiKP}}_{\text{obs}} - T^{\text{PcP}}_{\text{obs}}$ is the observed PKiKP-PcP traveltime residual, and $T^{\text{PKiKP}}_{\text{iasp91}} - T^{\text{PcP}}_{\text{iasp91}}$ is the predicted traveltime residual using the IASP91 model. Due to the different effects of Earth's ellipticity on the theoretical PKiKP and PcP traveltime, the PKiKP-PcP differential traveltime residuals in Supplementary Fig. 16 are calculated after corrections for ellipticity[53]. The corrected residuals are mostly negative in our study region, indicating a thinner liquid outer core. We also found that the PcP waves are generally slower than the theoretical arrival time (1–2 s), whereas the PKiKP waves are comparable to or faster than the theoretical travel time (Supplementary Fig. 3), which may suggest a deeper CMB than the IASP91 model in this region. Previous studies of the long-wavelength topography at CMB also demonstrated that the Earth's core radii are smaller than the global average under Central Asia and Siberia[54,55].

### Modeling of the ICB structure complexity
There have been successful applications of trial-and-error waveform modeling to quantify the fine-scale structure near the CMB and ICB. In contrast, grid search could provide the best model to explain those stacked PKiKP observations, but it requires a large number of forwarding computations[56]. When working in the frequency band of more than 2 Hz, the numerical algorithms are computationally costly while conducting a grid search. Because of high computational efficiency, the reflectivity method is suitable for conducting grid search at high frequencies. The seismogram $u(t)$ may be written as the convolution of the Earth response $G(t)$ with source time function s($t$):

$$u(t) = s(t) * G(t) \quad (2)$$

In this study, s($t$) is the PKiKP synthetic seismograms for the IASP91 model using the DSM, $G(t)$ is reflected pulses at interfaces, which depend on model and ray parameters. As the maximum separation between the two peaks of PKiKP is about 1 s, we explore parameter space with a step of 0.1 km for the thickness of layer variation from 1 to 8 km, a step of 5% for velocity, and density jumps from 5% to 70%. A grid search based on the reflectivity method for a range of thickness and seismic properties jump (velocity and density) combinations can be performed to find the optimal parameter combination, producing the higher cross-correlation coefficient between the synthetics and stacked PKiKP waveform (Supplementary Fig. 9). We then calculate the short period (with a dominant frequency of about 4 Hz) PKiKP synthetics with the accurate numerical algorithm DSM to fit those stacked PKiKP, based on those optimal parameter combinations in Supplementary Fig. 9.

## Data availability
The seimic data that support this study were accessed through the following data centers, Incorporated Research Institutions for

Seismology (IRIS) Data Management Center (http://service.iris.edu), Data Management Centre of China National Seismic Network (CNSN) at Institute of Geophysics, China Earthquake Administration. And data from CNSN in this study can be accessed at https://zenodo.org/record/8085938.

## Code availability

Seismic Analysis Code (SAC) and Generic Mapping Tools package (GMT)[57] are used for data processing and figure plotting. The open source numerical algorithm DSM[31] software was used to calculate the short-period PKiKP synthetics.

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

## Acknowledgements

This study is supported by National Natural Science Foundation of China (Grant No., 42030311, 42274077, 41925017), the National Key R&D Program of China (2021YFA0715100), the Science and Technology Innovation Talent Program of Hubei Province (2022EJD015), and the Fundamental Research Funds for the Central Universities (WK2080000144). We thank the Data Management Centre of China National Seismic Network at Institute of Geophysics, China Earthquake Administration, Incorporated Research Institutions for Seismology Data Management Center for providing data used in this study.

## Author contributions

S.N. and B.Z. conceived this study together. B.Z. performed seismic data processing, waveform modeling, analysis, figures preparation, and drafted the manuscript under the supervision of S.N. and W.B.W.; Z.S. and D.S. implemented the finite-difference P-SV code to compute the synthetic PKiKP waveforms for topography models. W.Z.W. and Z.Q.W. contributed to the discussion of mineral physics interpretations. All authors discussed the results and contributed to the manuscript preparation.

## Competing interests

The authors declare no competing interests.
