## [Peer Review File · Nature Communications]

Small-scale layered structures at the inner core boundaryREVIEWER COMMENTS

Reviewer #1 (Remarks to the Author):

The study presents interesting observations on the Asian seismic stations of complex seismic waves that reflect from the inner-core boundary (PKiKP waves). Assuming that there shouldn't be differences between the PKiKP waves that bounce off the inner-core boundary (ICB) and PcP waves that bounce off the core-mantle boundary (CMB), the authors argue that the precursors observed for the PKiKP waves are anomalous, i.e., not what we would expect if the Earth looked like the 1D models of the Earth suggest. They eliminate other possible causes and conclude that the waveform complexities they observe originate at the ICB. Through direct (forward) modeling, they further demonstrate that it is possible to simulate this complexity with the laminated structures near the ICB.

The methodology is not particularly innovative—the authors do what many others have done before for other structures within the Earth. The proposed Earth model is not unique, nor do the data require it, but it explains the observations. That is a subtle difference, which, in all honesty, permeates all geophysical studies, and thus the authors should not be penalized because of it. The deep Earth's observations and interpretations are intriguing and should interest the Nature group of journals. The paper is presented well overall but somewhat skewed towards specific papers irrelevant to this study and the readers' understanding of the problem. On the other hand, some relevant work done in the past is not referenced in the current paper, so the discussions and interpretations could be improved further in that direction. I commend the authors for using 2D numerical simulations to illustrate how different topographies may shape synthetic waveforms.

I support this study but want to see improvements by incorporating discussions about relevant past work and the required justifications that will strengthen the presentation and robustness of the reported work.

Main points:

- 1) Please show the source time functions (STF) for all four events used in the study. If they are unavailable, please include your own computations based on the P waves recorded between 30 and 90 degrees. Then, discuss how the observed PcP and PKiKP waves look compared to the STFs derived from the P waves. Is there a possibility that the STF is complex and that a PcP is affected by the passage through the lowermost mantle, whereas PKiKP was not? Please include a discussion on this.
- 2) Note that there are two highly relevant papers by the ANU group using very similar data regarding the PcP and PKiKP, the source location, and the recorded stations that the authors don't consider. Tkalcic, Kennett, Cormier (2009) in GJI and Tkalcic, Cormier, Kennett, He (2010) in PEPI study the variation of PKiKP waveforms in Asia using some of the same source-receiver geometries and find a lot of variation. Importantly, they demonstrate that, while the ICB displays different sharpness (the same conclusion you reached), some complexities and differences between the PKiKP and PcP can originate in the upper mantle and lithosphere. Please compare the maps of the ICB reflection points between your and those studies. From that point of view, your discussions and rationale in which you dismiss the mantle structure are incomplete and almost demand taking a closer look at those papers and making them a reference for your analysis, comparisons, and discussions.
- 3) Please incorporate the list of stations used in your study through additional tables (as a part of the supplementary material) and show the PcP/PKiKP amplitude ratios and the nature of PKiKP in your table (would be easier to follow). What is the correlation between the amplitude ratios and structures modeled in your synthetic experiments?
- 4) Many references used in the introduction don't appear particularly relevant, and they were not used in the discussion part. Some of them could be omitted and replaced by more relevant references for this work, e.g., References: 12, 27, 29, 30, 32, 37, and possibly others.
- 5) Figure 2 deserves more attention, particularly regarding the theoretical slowness of PKiKP and

PcP vs. the observed slowness. Please comment on the observed differences between PcP and PKiKP slowness, and show the theoretical estimates in your diagrams. Please show this for all four events, not just 2 of them, perhaps in the supplementary. Have you computed the beams and what do they say about your observations?

Minor points:

Line 63: no capitalization.

Figure 1: The lines in my manuscript version are orange, not red.

Line 76: explain the S/N ratio and what it means. Does it refer to PKiKP or PcP, or both?

Figure 3: explain what you mean by "reflected points." It would be better to say "the ICB reflection" points in high-frequency approximation...

Line 145: explain the rationale behind using IASP91 instead of ak135, or even newer, ek137

Comment on the distance at the CMB in the same way you did when talking about the separation at the 660 discontinuity

Line 235: there is a typo; geodynamicist  geodynamicists

Reference 26: there is a typo; Engdah  Engdahl

Reference 35: Earth ' s  Earth's

Reviewer #2 (Remarks to the Author):

This is very interesting research. However, more complete explanation of the process of identifying the four events presented is necessary before I can assess the interpretations and significance of the results.

The text mentions quite a few arrays, some with decades of recording, but presents the results for just four events, without describing how they were selected. In contrast, ref 18 by Tian and Wen (2017) gives a lengthy description of the analysis of 1000 events chosen by an easily understood process, and includes the criteria for finding the only 11 events among them with similar and compact PcP and PKiKP, which allows judgment of the overall database from which the observations were selected.

Without such information it is hard to understand the constraints on lateral variation, and the possibility of multipathing due to boundary undulations or mantle features such as subducted slabs or CMB features. Very few observations of PKiKP waveform distortion due to ICB fine-scale layering have been found, so exploration of alternatives has a higher hurdle than usual.

I haven't read the rest of the paper carefully, and am optimistic this paper has the strong possibility of being a valuable contribution once the data and the selection process are explained.

NOTE: The reviewers' comments and suggestions are in black text, while our point-by-point responses can be found in blue text.

REVIEWER COMMENTS

Reviewer #1 (Remarks to the Author):

The study presents interesting observations on the Asian seismic stations of complex seismic waves that reflect from the inner-core boundary (PKiKP waves). Assuming that there shouldn't be differences between the PKiKP waves that bounce off the inner-core boundary (ICB) and PcP waves that bounce off the core-mantle boundary (CMB), the authors argue that the precursors observed for the PKiKP waves are anomalous, i.e., not what we would expect if the Earth looked like the 1D models of the Earth suggest. They eliminate other possible causes and conclude that the waveform complexities they observe originate at the ICB. Through direct (forward) modeling, they further demonstrate that it is possible to simulate this complexity with the laminated structures near the ICB.

The methodology is not particularly innovative—the authors do what many others have done before for other structures within the Earth. The proposed Earth model is not unique, nor do the data require it, but it explains the observations. That is a subtle difference, which, in all honesty, permeates all geophysical studies, and thus the authors should not be penalized because of it. The deep Earth's observations and interpretations are intriguing and should interest the Nature group of journals. The paper is presented well overall but somewhat skewed towards specific papers irrelevant to this study and the readers' understanding of the problem. On the other hand, some relevant work done in the past is not referenced in the current paper, so the discussions and interpretations could be improved further in that direction. I commend the authors for using 2D numerical simulations to illustrate how different topographies may shape synthetic waveforms.

I support this study but want to see improvements by incorporating discussions about relevant past work and the required justifications that will strengthen the presentation and robustness of the reported work.

Reply: We appreciate your insightful and constructive comments and suggestions, which greatly

improved the manuscript. We have addressed these concerns and made a revision of manuscript. According to the suggestions, we have:

(1) Shown the source time function (STF) for the four events and compared STF with PcP and PKiKP observations, then included a few sentences of discussion on this in the updated paragraph 12.

(2) Compared the maps of the ICB reflection points between our and previous studies, and presented analysis, comparisons and discussions in the updated paragraph 13.

(3) Calculated PKiKP/PcP amplitudes and added the discussion of the correlation between the amplitude ratios and structural models in the revised manuscript (updated paragraph 14).

(4) Cited more relevant references in the introduction and discussion sections. Strengthened the presentation and robustness of our work by incorporating discussions about relevant previous studies and the required justifications.

(5) Calculated vespagrams (in the updated Supplementary Fig. 2.) and beams (in the updated Supplementary Fig. 3.) for these seismic array observations from all the four Events using the Nth-root process and frequency-wavenumber analysis technique, respectively.

Main points:

1) Please show the source time functions (STF) for all four events used in the study. If they are unavailable, please include your own computations based on the P waves recorded between 30 and 90 degrees. Then, discuss how the observed PcP and PKiKP waves look compared to the STFs derived from the P waves. Is there a possibility that the STF is complex and that a PcP is affected by the passage through the lowermost mantle, whereas PKiKP was not? Please include a discussion on this.

Reply: Thanks for your suggestions and questions. We have accordingly shown the source time functions for all four events used in supplementary Fig. 12. The STFs of Event 1 and Event 3 were

downloaded from the SCARDEC source time functions database (Vallee & Douet, 2016), which provide broadband STFs with the SCARDEC method (used the teleseismic body waves). But this database does not provide the STFs of Events 2 and 4. Thus we stacked the broadband displacement seismograms of direct P-waves in the distance range of 40-60 degrees to obtain their empirical STFs. Then we compared the observed PcP and PKiKP waveforms with the STF derived from the P waves in the updated Supplementary Fig. 13. Indeed, the source time functions (STFs) of Events 3-4 exhibit greater complexity compared to their corresponding observed PcP and PKiKP waves. This complexity is attributed to the epicentral distances of the PcP and PKiKP observations utilized in this study being less than 44° . As a result, there is a notable difference between the take-off angles of the direct P wave and those of PcP and PKiKP, as depicted in Supplementary Fig. 13a. Moreover, PcP may be affected by anomalous structures near the CMB, leading to a more complex PcP waveform compared to PKiKP. For instance, the PcP waveforms recorded by the Makanchi array at Kazakhstan for Event 4 exhibit significantly greater complexity compared to the corresponding PKiKP waveforms in the frequency band of 1.5-3.0 Hz in Supplementary Fig. 13b. To effectively demonstrate the complexity of PKiKP waveforms, it is preferable for the waveform of the reference phase PcP to show a comparatively simple feature. Consequently, we have excluded the observation data characterized by more intricate PcP waveforms.

And we have discussed these contents in the updated paragraph 12 of the revised manuscript, as follows:

“Complexity of seismic source might hinder reliable modeling of PcP and PKiKP. Thus the source time function (STF) is important for analyzing waveform anomalies of PcP and PKiKP. According to Supplementary Fig. 12, Event 1 displays a simple STF lasting around 1 second. Despite Event 2 having a magnitude of $M6.7$, its STF duration was only about 2 s, indicating a relatively simple source time function, possibly due to it being a deep earthquake with supershear rupture characteristics. Event 3 lasted approximately 2.8 s, but its shape of STF differs from Events 1 and 2, with a delayed peak. Event 4 is a nuclear explosion source with a simplicity of the source physics. However, due to the shallow source depth, the surface reflected wave pP will affect the direct P wave, resulting in a negative plue in the STF. The observed PcP waves of Events 1 and 2 in Figure 3b,c (marked as Northeast Asia and Central Asia) are similar to the duration time of STFs. The PKiKP

waveform is much more complex than the PcP waveform and its STF. However, the source time functions (STFs) of Events 3-4 exhibit obvious complexity compared to their corresponding observed PcP and PKiKP waves. This complexity may be attributed to the epicentral distances of the PcP and PKiKP observations utilized in this study being less than 44° . Consequently, a noticeable disparity exists between the take-off angles of the direct P wave and those of PcP and PKiKP, as illustrated in Supplementary Fig. 13a. Moreover, PcP may be affected by anomalous structures near the CMB, leading to a more complex PcP waveform compared to PKiKP. For instance, the PcP waveforms recorded by the Makanchi array at Kazakhstan for Event 4 exhibit significantly greater complexity compared to the corresponding PKiKP waveforms in the frequency band of 1.5-3.0 Hz in Supplementary Fig. 13b. To effectively observe the complexity of PKiKP waveforms, it is desirable for the reference phase PcP waveform to exhibit a relatively simple feature. Consequently, we have excluded observation data characterized by more intricate PcP waveforms.”

Supplementary Fig. 12. The source time functions (STF) for all four events used in the study.

The STFs of Event 1 and Events 3 were downloaded from SCARDEC source time functions database (<http://scardec.projects.sismo.ipgp.fr/>), which provide broadband STFs with the SCARDEC method (Vallee & Douet, 2016). We stacked the broadband displacement seismograms

of direct P-waves in the distance range of 40-60 degrees to obtain empirical STF of Event 2 and Events 4.

Supplementary Fig. 13. (a) Seismic raypaths of P, PcP and PKiKP phases. (b) Comparisons of the observed PKiKP (orange) and PcP (blue) waveforms at Makanchi array in Kazakhstan for Event 4 with its STF (red) derived from the P waves.

References:

Vallée, M. & Douet, V. A new database of source time functions (STFs) extracted from the SCARDEC method. *Phys. Earth Planet. Inter.* 257, 149–157 (2016).

2) Note that there are two highly relevant papers by the ANU group using very similar data regarding the PcP and PKiKP, the source location, and the recorded stations that the authors don't consider. Tkalčić, Kennett, Cormier (2009) in GJI and Tkalčić, Cormier, Kennett, He (2010) in PEPI study the variation of PKiKP waveforms in Asia using some of the same source-receiver geometries and find a lot of variation. Importantly, they demonstrate that, while the ICB displays different sharpness

(the same conclusion you reached), some complexities and differences between the PKiKP and PcP can originate in the upper mantle and lithosphere. Please compare the maps of the ICB reflection points between your and those studies. From that point of view, your discussions and rationale in which you dismiss the mantle structure are incomplete and almost demand taking a closer look at those papers and making them a reference for your analysis, comparisons, and discussions.

Reply: Indeed, these two papers are indeed highly relevant to our research work, both of us used the pre-critical PKiKP and PcP observations, and some sampling points are located near our study area in Central Asia. They calculated an abundant record of PcP and PKiKP travel times and amplitudes from a single earthquake and a nuclear explosion, quantitatively analyzing the influence of heterogeneities near the surface and CMB regions on the amplitude ratio of PKiKP/PcP, and found some complexities and differences between the PKiKP and PcP in individual seismograms can originate in the upper mantle and lithosphere (Tkalčić et al., 2009, 2010).

Following your suggestions, we have compared the maps of the ICB reflection points between our and those studies in the Supplementary Fig. 14a. Although there is no direct overlap of ICB reflection points between their and our studies, our observations of PcP and PKiKP might still be influenced by similar shallow structures due to the proximity of some of our station at Central Asia. Previous seismological studies demonstrated that the array stacking technique could effectively enhance the SNR of core phases such as PcP and PKiKP while suppressing scattering waves from strong small-scale heterogeneities near the surface (Rost & Thomas, 2002; Tian & Wen, 2017). There has been an increasing accumulation of seismic array observational data in recent years, which has provided a solid foundation for the application of station stacking techniques to image the fine-scale structure of the inner core (Tkalčić, 2015). Therefore, to reduce the effects of near-surface structure, we have stacked these array PcP and PKiKP data based on the time-frequency domain phase-weighted stack (tf-PWS) method in our study.

We have accordingly added the discussion of the effects of the mantle structure in our revised manuscript in the updated paragraph 13, as follows:

“Earth’s near-surface layers including the lithosphere and upper mantle have a strong small-scale heterogeneity⁴⁰, which may affect the amplitude and waveforms of PKiKP and PcP. For instance,

Tkalčić et al., (2009, 2010) calculated an abundant record of PcP and PKiKP travel times and amplitudes from a single earthquake and a nuclear explosion, quantitatively analyzing the influence of small-scale heterogeneities near the surface regions on the amplitude ratio of PKiKP/PcP. They found some complexities and differences between the PKiKP and PcP in individual seismograms can originate in the upper mantle and lithosphere. Although there is no direct overlap of ICB reflection points between their and our studies, our observations of PcP and PKiKP might still be influenced by similar shallow structures due to the proximity of some of our stations in Central Asia (Supplementary Fig. 14a). Previous seismological studies demonstrated that the array stacking technique could effectively enhance the SNR of core phases such as PcP and PKiKP while suppressing scattering waves from strong small-scale heterogeneities near the surface^{22,23,28}. There has been an increasing accumulation of seismic array observational data in recent years, which has provided a solid foundation for the application of station stacking techniques to image the fine-scale structure of the inner core¹¹. Therefore, to reduce the effects of near-surface structure, we have stacked these array PcP and PKiKP data based on the time-frequency domain phase-weighted stack (tf-PWS) method in our study.”

References:

- Tkalčić, H. Complex inner core of the Earth : The last frontier. *Rev. Geophys.* 53, 59–94 (2015).
- Tkalčić, H., Kennett, B. L. N. & Cormier, V. F. On the inner – outer core density contrast from PKiKP / PcP amplitude ratios and uncertainties caused by seismic noise. *Geophys. J. Int.* 179, 425–443 (2009).
- Krasnoshchekov, D. N., Kaazik, P. B. & Ovtchinnikov, V. M. Seismological evidence for mosaic structure of the surface of the Earth’s inner core. *Nature* 435, 483–487 (2005).
- Tian, D. & Wen, L. Seismological evidence for a localized mushy zone at the Earth’s inner core boundary. *Nat. Commun.* 8, 165 (2017).
- Rost, S. & Thomas, C. Array seismology: Methods and applications. *Rev. Geophys.* 40, 1008 (2002).
- Tkalčić, H., Cormier, V. F., Kennett, B. L. N. & He, K. Steep reflections from the earth’s core reveal small-scale heterogeneity in the upper mantle. *Phys. Earth Planet. Inter.* 178, 80–91 (2010).

Shearer, P. M. Deep Earth Structure - Seismic Scattering in the Deep Earth. Treatise on Geophysics
vol. 1 (Elsevier B.V., 2015).

3) Please incorporate the list of stations used in your study through additional tables (as a part of the supplementary material) and show the PcP/PKiKP amplitude ratios and the nature of PKiKP in your table (would be easier to follow). What is the correlation between the amplitude ratios and structures modeled in your synthetic experiments?

Reply: Thanks for your valuable suggestions and question. Considering that we have selected over 400 high-quality PcP and PKiKP waveform pairs, it is quite challenging to present such a large amount of station information through tables in the Word document. Accordingly, we have incorporated the list of stations, the corresponding array, SNR of PcP and PKiKP, and the PKiKP/PcP amplitude ratios in a text file with the suffix “.txt”, which is also included as part of the supplementary material. Moreover, we have revised the Supplementary Table 1 by adding the number of PcP and PKiKP traces. Furthermore, we have plotted the PKiKP/PcP amplitude ratios measured from each trace, as well as the theoretical amplitude ratios calculated from the IASP91 model in the supplementary Figure 14b for better clarity and understanding.

Theoretically, the PKiKP/PcP ratios observed in the regions where the ICB exhibits small-scale layered structures should be smaller than the theoretical prediction from the IASP91 model. From Supplementary Fig. 14b, it is indeed evident that the amplitude ratios of PKiKP/PcP in the Central Asia region (IA1), Siberia (IA4 and IA5), and partial of Northeast Asia (IA6) are significantly smaller than the theoretical values of the IASP91 model. And the regions such as Mongolia and most of Northeast Asia where we consider the ICB to be normal, the amplitude ratio of PKiKP/PcP is comparable to the theoretical values. However, the ICB abnormal regions such as IA2 and IA3 in Central Asia, their PKiKP/PcP amplitude ratios are larger than the theoretical value. Because the SNR of PcP phases is relatively low in this region, with only a few events from Hindukush being observable PcP signals at KURA and BVA array. Conversely, the SNR ratio for PKiKP is comparatively good, as clear PKiKP seismic phases were observed in over ten events.

We have accordingly added the discussion of the correlation between the amplitude ratios and

structural models in the revised manuscript in the updated paragraph 14, as follows:

“The amplitude ratios of PKiKP and PcP in time domain are widely used to constrain ICB properties^{16–18,21}. However, significant variations in the amplitude ratios of PKiKP/PcP are observed across different stations, including within the small-aperture dense arrays (Supplementary Fig. 14b). These variations may stem from the influence of multi-scale heterogeneities along the propagation paths of PKiKP and PcP, as well as disparities in instrument performance and installation conditions. Theoretically, in regions where the ICB exhibits small-scale layered structures, the observed PKiKP/PcP ratios should be smaller than the theoretical predictions from the IASP91 model. From Supplementary Fig. 14b, it is indeed evident that the amplitude ratios of PKiKP/PcP in the Central Asia region (IA1), Siberia (IA4 and IA5), and partial of Northeast Asia (IA6) are significantly smaller than the theoretical values of the IASP91 model. In regions such as Mongolia and most of Northeast Asia, where the ICB is considered to be normal, the amplitude ratio of PKiKP/PcP is comparable to the theoretical values. However, in the anomalous ICB regions such as IA2 and IA3 beneath Central Asia, their PKiKP/PcP amplitude ratios are larger than the theoretical value. Because the SNR of PcP phases is relatively low in this region, only a few events from Hindukush show observable PcP signals at KURA and BVA array. In contrast, the SNR ratio for PKiKP is relatively good, as clear PKiKP seismic phases were observed for more than ten events. Therefore, the combined effects of multi-scale heterogeneities near the lithosphere, upper mantle, and the CMB, as well as contamination from S-coda waves, have a noticeable impact on the PcP phase in the Central Asia region.”

Supplementary Fig. 14. (a) ICB reflection points distribution in this and some previous studies beneath the Central and East Asia. (b) Measured PKiKP/PcP amplitude ratios from observations and synthetics in the frequency band of 2-3 Hz. Colored crosses denote the normal ICB regions. Whereas red circles denote the ICB abnormal regions in this study. Squares denote the regions where anomalous amplitudes or waveforms of PKiKP have been observed in previous studies. Blue line is the amplitude ratio of PKiKP to PcP that obtained from synthetic seismograms for the IASP91 model.

References:

Tanaka, S. & Tkalčić, H. Complex inner core boundary from frequency characteristics of the

reflection coefficients of PKiKP waves observed by Hi-net. *Progress in Earth and Planetary Science* vol. 2 1–16 (2015).

Shearer, P. M. & Masters, G. G. The density and shear velocity contrast at the inner core boundary. *Geophys. J. Int.* 102, 491–498 (1990).

Shen, Z., Ai, Y., He, Y. & Jiang, M. Using pre-critical PKiKP-PcP phases to constrain the regional structures of the inner core boundary beneath East Asia. *Phys. Earth Planet. Inter.* 252, 37–48 (2016).

Tkalčić, H., Kennett, B. L. N. & Cormier, V. F. On the inner – outer core density contrast from PKiKP / PcP amplitude ratios and uncertainties caused by seismic noise. *Geophys. J. Int.* 179, 425–443 (2009).

4) Many references used in the introduction don't appear particularly relevant, and they were not used in the discussion part. Some of them could be omitted and replaced by more relevant references for this work, e.g., References: 12, 27, 29, 30, 32, 37, and possibly others.

Reply: Thanks for your comments and suggestions. Following your suggestions, we have replaced them with more relevant references and incorporated further relevant discussions into the paper. For example, we have cited more seismological studies of the ICB in the introduction and discussion, such as Tkalčić (2015, *Rev. Geophys.*), Wen (2006, *Science*), Cormier (2007, *EPSL*), Tkalčić et al., (2009, *GJI*; 2010, *PEPI*), Koper & Dombrovskaya (2005, *EPSL*), Shearer & Master (1990, *GJI*), etc. Therefore, more than 60% of the references cited in the introduction section were utilized in the discussion part as well in the revised manuscript.

5) Figure 2 deserves more attention, particularly regarding the theoretical slowness of PKiKP and PcP vs. the observed slowness. Please comment on the observed differences between PcP and PKiKP slowness, and show the theoretical estimates in your diagrams. Please show this for all four events, not just 2 of them, perhaps in the supplementary. Have you computed the beams and what do they say about your observations?

Reply: Thanks for your suggestions and question. The analysis of slowness and back-azimuth is important for confirming the robustness of the selected PcP and PKiKP phases. Following your suggestions, we have respectively calculated vespagrams (in the updated Supplementary Fig. 2.) and beams (in the updated Supplementary Fig. 3.) for these seismic array observations from all four Events using the Nth-root process and frequency-wavenumber analysis technique. Furthermore, we have also accordingly provided clarifications on the observed differences in slowness and back-azimuth of PcP and PKiKP between the theoretical estimates and the measurements in lines 97-105 in the revised manuscript.

Figure 2 displays the spectrograms of stacked PcP and PKiKP at different seismic arrays. It illustrates the changes of the PcP and PKiKP seismic phases in the time-frequency domain, which is also a crucial step in obtaining the stacked waveforms shown in Figures 3b and 3c. To better show the features of PcP and PKiKP energy at different frequencies, we aligned them based on the hand-picked travel times. Indeed, only showing the spectrograms for two events is insufficient, following your suggestion, we have included the spectrograms of stacked PcP and PKiKP at the KURA array from Events 3 and 4 in the updated Supplementary Fig. 5.

Minor points:

Line 63: no capitalization.

Reply: Thanks. We have accordingly changed “Robust” to “robust”.

Figure 1: The lines in my manuscript version are orange, not red.

Reply: Sorry. We have accordingly changed “red” to “orange”.

Line 76: explain the S/N ratio and what it means. Does it refer to PKiKP or PcP, or both?

Reply: Thanks for your question. The signal-to-noise ratio (SNR) in this study is defined as the ratio of the peak amplitude of PcP or PKiKP to the ambient noise amplitude of waveform window 5 to 10 s before the PcP or PKiKP arrival. We have accordingly added the explanation in lines 82-85 in the revised manuscript.

Figure 3: explain what you mean by “reflected points.” It would be better to say “the ICB reflection” points in high-frequency approximation...

Reply: Thanks for your suggestion. Indeed, our expression “reflected points” was not accurate enough. I have revised it according to your suggestion, changing “reflected points” to “the ICB reflections” in the Figure 3 caption.

Line 145: explain the rationale behind using IASP91 instead of ak135, or even newer, ek137

Reply: Thanks for your questions. Previous studies on the structure of the deep Earth commonly used 1D reference models, including PREM, IASP91, AK135, and EK137 models, and so on. We mainly adopted the direct solution method (DSM) to calculate the short-period PcP and PKiKP synthetic seismograms in our study, and the model configuration in the DSM is represented in polynomial form. Kennett et al., (1991) provide multiple polynomial parameterizations for the IASP91 models. There are some differences in the velocity gradient at ICB between IASP91 and AK135, which may cause the difference in amplitude of PKiKP, but we mainly used information of waveforms of PKiKP in this study, therefore, we believe that all these models should be applicable.

Comment on the distance at the CMB in the same way you did when talking about the separation at the 660 discontinuity.

Reply: Thanks for your suggestion. We have accordingly commented on the separation distance between PcP and PKiKP sampling point at the CMB in this study, and added the information in lines 194-196 in the revised manuscript, as follow:

“The ray paths of PcP and PKiKP deviate substantially in the lower mantle, especially at the bottom of the mantle, spatial separation between PcP and PKiKP sampling point at the CMB is in the range of ~ 2.5° to 17.2°...”

Line 235: there is a typo; geodynamicist  geodynamicists

Reply: Thanks for pointing out the typo. We have accordingly revised “geodynamicist” to “geodynamicists”.

Reference 26: there is a typo; Engdah  Engdahl

Reply: Thanks for pointing out the typo. We have accordingly revised “Engdah” to “Engdahl”.

Reference 35: Earth ' s  Earth's

Reply: Thanks for pointing out the typo. We have accordingly revised “Earth ’ s” to “Earth’s”.

Reviewer #2 (Remarks to the Author):

This is very interesting research. However, more complete explanation of the process of identifying the four events presented is necessary before I can assess the interpretations and significance of the results.

Reply: Thanks for your comments and suggestions. Indeed, the complete explanation of the identifying process of the four events is necessary, and the description is insufficient in the original manuscript. Following your suggestions, we have made more complete explanation of the process of identifying the four events presented in the updated paragraph 3,4, and 5 in the revised manuscript.

The text mentions quite a few arrays, some with decades of recording, but presents the results for just four events, without describing how they were selected. In contrast, ref 18 by Tian and Wen (2017) gives a lengthy description of the analysis of 1000 events chosen by an easily understood process, and includes the criteria for finding the only 11 events among them with similar and compact PcP and PKiKP, which allows judgment of the overall database from which the observations were selected.

Without such information it is hard to understand the constraints on lateral variation, and the possibility of multipathing due to boundary undulations or mantle features such as subducted slabs or CMB features. Very few observations of PKiKP waveform distortion due to ICB fine-scale layering have been found, so exploration of alternatives has a higher hurdle than usual.

I haven't read the rest of the paper carefully, and am optimistic this paper has the strong possibility of being a valuable contribution once the data and the selection process are explained.

Reply: Thanks for your comments. Follow your suggestion, we have rephrased the content in the PKiKP Observations section, provided a more comprehensive description of the data selection process in the updated paragraph 3, 4, and 5 in the revised manuscript. Indeed, selecting an adequate number of PcP and PKiKP seismic phase pairs with high quality has always been crucial for the study of the fine structure of the ICB, as well as a significant challenge. This is because of pre-critical PKiKP is weak at short epicentral distance, and may be contaminated by other high-

frequency signals with similar travel times. For instance, Tian and Wen (2017) found 11 events with high-quality PcP and PKiKP pairs from over 1100 seismic events with depths greater than 30km. And Shearer and Masters (1990) also found that only ~5% of the vertical-component seismograms in the epicentral distance 20-90° exhibit observable PKiKP phases.

Tian and Wen (2017) have provided a detailed investigation on the ICB structure beneath the west Pacific, so we primarily focus on the fine structure of the ICB beneath Central Asia, Siberia and Northeast Asia. To exclude potential contaminations on the PKiKP and PcP due to the earthquake rupture complexity and lithospheric scattering, we carefully scrutinize all the seismic recordings of moderate events (~M6) deeper than 80 km. This is because of strong earthquakes have complicated source time function and small events might be too weak to excite observable PcP and PKiKP waves. There are more than 440 events (including 438 deep earthquakes and 6 nuclear explosions) from January 1994 to December 2021 in our study regions, which are recorded by those seismic arrays within the epicentral distances of 50°. These seismic arrays primarily consist of broadband (BH?), high broadband (HH?), and short-period (SH?) seismometers. These earthquakes are mostly distributed in the Hindukush, Myanmar, and the western Pacific subduction zone regions.

1. To choose high-quality PcP and PKiKP phases, we only retain waveforms with a signal-to-noise ratio (SNR) ≥ 2.0 , SNR is defined as the ratio of the peak amplitude of PcP or PKiKP to the ambient noise amplitude of waveform window 5 to 10 s before the PcP or PKiKP arrival. We found that high noise levels impede the observation of PKiKP signals in majority of these seismograms. For events with high SNRs in their PcP and PKiKP seismograms, we additionally performed manual inspections to assess their quality. For valuable small-aperture dense array observations, we retain any visually identifiable PcP and PKiKP signals, even if their signal-to-noise ratio is less than 2. This is because by examining the waveforms within an array, we can exclude most random process in individual seismograms.

2. The slowness and backazimuth analysis are helpful in confirming the robustness of selected PcP and PKiKP phases. Therefore, we firstly compute the vespagrams for these seismic arrays using a nonlinear stacking method (Nth-root process) and the slowness of the observed signals that are similar to the theoretical predictions of PcP and PKiKP (Supplementary Fig. 3). Furthermore, we determine the slowness and backazimuth of the observed signals at different seismic arrays with the

frequency-wavenumber analysis technique.

3. Moreover, other seismic phases of different propagation paths may have travel times close to those of the PKiKP or PcP at some epicentral distances, such as S waves (Supplementary Fig. 2). Although we could suppress those interfering signals by bandpass filtering or array stacking technologies, the contaminated PKiKP waveforms are difficult to be fully recovered and could easily be misinterpreted as the result of small-scale heterogeneity in the inner core surface. Therefore, seismic observations of PKiKP phase were excluded for analysis when it is near the interfering waves such as S and ScS. For example, in the Hindu Kush region, there are many moderate events with a depth of around 200 km, and an abundance of high-quality PKiKP waveforms have been recorded at those dense arrays in Central Asia. But the reference phase PcP was severely contaminated by the strong S-coda waves. In this region, good records of PKiKP and PcP phases are simultaneously observed only for Event 1, while other events show contaminated PcP waves. Along the ray path from Myanmar to Central Asia, the epicentral distance is approximately 30° , and the PKiKP waves are heavily affected by the high-frequency ScS waveforms (Supplementary Fig. 2a).

After the screening criteria mentioned above, we have finally retained waveform data from four events, collected more than 400 high-quality waveforms with both clear PcP and PKiKP phases in the frequency range of 2-3 Hz, which mainly sampled the ICB beneath Central Asia, Siberia, and Northeast Asia. We have incorporated the list of stations, the corresponding array, SNR of PcP and PKiKP, and the PKiKP/PcP amplitude ratios in a text file with the suffix “.txt”, which is also included as part of the supplementary material. Moreover, we have revised the Supplementary Table 1 by adding the number of PcP and PKiKP pairs for each event.

REVIEWER COMMENTS

Reviewer #1 (Remarks to the Author):

First of all, I commend the authors for a thorough revision, and for not dodging any bullets in the two reviews. As a result, I believe the manuscript is now in much better shape. It was interesting to see the STF analyses and how complexities can affect the interpretations.

Some sections are heavily reworked especially the discussion section. The authors recognise that some waveform complexities and interpretations may come from data processing and also originate elsewhere from Earth's interior. Therefore, additional minor changes in the summary and the conclusions to reflect these findings would serve the authors well in conveying the message to the broader audience. I would suggest that the authors add some extra clauses in the existing summary and importantly, in the conclusion section accordingly.

Also, I think that the phrase "Overall, we provide seismological evidence for a laminated ICB..." in the conclusions is unnecessarily too strong. It would be wiser to tone down this statement, perhaps by saying something along these lines: "Overall, we model our observations with a laminated ICB..." or something along these lines.

Finally, it would be interesting to say something about how your model of a laminated ICB would affect the heat flow from the inner core for the most recent conductivity estimates. I leave it to the authors to decide if necessary to comment, but it would perhaps help attract a wider readership.

Again, thanks for adopting my recommendations, and I hope the manuscript is widely read and valued by the community.

Reviewer #2 (Remarks to the Author):

The authors have provided much more detail. It is still very hard to read the paper, but I did make it through this iteration. I commend the other reviewer for their thorough initial review.

These observations are well made and significant, so I am optimistic this paper has good potential. However, Event 1 is the only observation of a small, clear precursor to PKiKP 1s ahead of which I'm aware, so more exploration should be undertaken to rule out alternative explanations and the presentation needs a careful re-writing and proof reading.

While the modeling is impressively comprehensive, in the end I mostly take away the stated conclusions "PKiKP distortions are mostly caused by the seismic anomalies near the ICB" and "the distortion .. may be the combined effect of laminated and rough ICB" (and I'd add lateral variation in the uppermost IC). It is very difficult to interpret layering rather than more disordered structure from these observations as presented.

The authors could save the reader a lot of trouble by summarizing the observations at the start. Essentially, they look at four earthquakes (3 + an explosion), comparing PKiKP to PcP - one in the west shows a small 1s precursor to PKiKP on three tight arrays spread across a few hundred km, several(?) events in the east either show a simple ICB or have PKiKP with a few extra cycles on the end in the passband above 2 Hz. It literally took me hours, between the figures, text, and supplement, to think I knew how the observations are being interpreted. It would be helpful, for example, to label figures with which event and which array is shown in each frame.

The data surveyed remains unclear. It says "All 400+ events greater than M6.0 from 2012 to 2016" (on what magnitude scales, in what area?) were examined (on all arrays?). But events 1 & 3 were 5.8, less than 6, and no area is defined. The text says all data screened with requirement SNR > 2, next paragraph implies any visible arrival was retained, but for what data? Then it is noted "Found two additional events" - why were these not in the original collection? I'm still left

with the impression the authors found 2 interesting events, then when pressed found two more, rather than a more systematic survey, which is ok, but should be accurately described.

I see reference to phase-weighted stacks and nth-root stacks. Both distort waveforms, precluding estimation of the amplitude of weaker arrivals and thus comparison with synthetics. I'd greatly prefer to see linear stacks, which would preserve waveform and frequency content.

Figs 3b and 3c show very similar arrivals for all three earthquakes (I think Event 4 waveforms are only ever shown in Fig S13). These M6 events are nearly identical? The source time functions estimated in Fig S12 differs between events and in several cases is longer than the PKiKP and PcP arrivals. The arrival is somehow deconvolved? I strongly suspect that the events chosen here for sharp arrivals (and in Tian and Wen's paper) are the small fraction that send a strong directivity pulse straight down to the CMB and ICB, and only a portion of the source time function is appearing. This case would not negate the conclusions, but might make the waveforms more sensitive to the take-off angle and azimuth, giving more possibilities to explain the observed differences and pose more difficulties interpreting the absolute and relative amplitudes.

Only event 1 has the 1s precursor, it's on all 3 western arrays. Is there no second event to check in decades of recordings? PcP doesn't have to be clean to see if PKiKP often has the precursor. One could also examine the direct mantle P waves to assess the presence or absence of a weak beginning 1s before a strong pulse in the source. For that matter, I'd like to see the mantle waves for Event 1, which has the most anomalous observations, and the P wave should be presented in the same passband, 1-4 Hz, as the ICB reflections.

Minor consideration:

Summing over the main array should average out lateral heterogeneity, producing a simpler apparent arrival in the stack, for example in IN3 - do stacks of PKiKP on smaller subarrays have more complex waveforms?

**NOTE: The reviewers' comments and suggestions are in black text, while our**
**point-by-point responses can be found in blue text.**

REVIEWER COMMENTS

Reviewer #1 (Remarks to the Author):

First of all, I commend the authors for a thorough revision, and for not dodging any bullets in the
two reviews. As a result, I believe the manuscript is now in much better shape. It was interesting to
see the STF analyses and how complexities can affect the interpretations.

Reply: Thanks for your encouragement. We appreciate your thorough and helpful review. Your
insightful and constructive comments and suggestions have greatly improved the manuscript.

Some sections are heavily reworked especially the discussion section. The authors recognise that
some waveform complexities and interpretations may come from data processing and also originate
elsewhere from Earth's interior. Therefore, additional minor changes in the summary and the
conclusions to reflect these findings would serve the authors well in conveying the message to the
broader audience. I would suggest that the authors add some extra clauses in the existing summary
and importantly, in the conclusion section accordingly.

Reply: Thanks for your comments and suggestions. We have accordingly added additional changes
in the existing Abstract and Conclusion sections, as follows:

*"Here, we utilize a new dataset of pre-critical PKiKP waveforms to constrain the fine structure at*
*the ICB, considering the influence of various factors such as source complexity, structural*
*anomalies in the mantle, and properties at the ICB."*

Also, I think that the phrase "Overall, we provide seismological evidence for a laminated ICB..." in
the conclusions is unnecessarily too strong. It would be wiser to tone down this statement, perhaps
by saying something along these lines: "Overall, we model our observations with a laminated ICB..."
or something along these lines.

Reply: Thanks for your suggestion. We have rephrased the sentence in the conclusion to tone down
this statement, as follows:

*"Our modeling suggests a sharp ICB beneath Mongolia and most of Northeast Asia, but a locally*
*laminated ICB structure beneath Central Asia, Siberia, and part of Northeast Asia. The complex*
*ICB structure might be explained by either the existence of a kilometer-scale thickness of mushy*
*zone, or the localized coexistence of bcc and hcp iron phase at the ICB."*

Finally, it would be interesting to say something about how your model of a laminated ICB would
affect the heat flow from the inner core for the most recent conductivity estimates. I leave it to the
authors to decide if necessary to comment, but it would perhaps help attract a wider readership.

Reply: Thanks for your comment. The heat flow from the inner core could potentially be subject to
local influences stemming from the laminated ICB structure. Accurate estimation of this heat flow

necessitates knowledge of the thermal conductivity of hcp and bcc iron alloys under ICB conditions.
However, there is a scarcity of data in the literature concerning both the hcp and bcc phases under
ICB conditions at present.

We have accordingly incorporated the discussion on the influence of a laminated ICB model on the
heat flow from the inner core for the most recent conductivity estimates in the discussion section.

Again, thanks for adopting my recommendations, and I hope the manuscript is widely read and
valued by the community.

Reply: Thanks very much.

Reviewer #2 (Remarks to the Author):

**Comment 1:**

The authors have provided much more detail. It is still very hard to read the paper, but I did make it
through this iteration. I commend the other reviewer for their thorough initial review. These
observations are well made and significant, so I am optimistic this paper has good potential.
However, Event 1 is the only observation of a small, clear precursor to PKiKP 1s ahead of which
I'm aware, so more exploration should be undertaken to rule out alternative explanations and the
presentation needs a careful re-writing and proof reading.

Reply: Thanks for your thorough review of the manuscript. We appreciate your constructive
suggestions and questions, which have greatly improved the manuscript. To enhance the readability
of the manuscript, we have accordingly polished and restructured the revised manuscript.

Following your suggestion, we have accordingly conducted a comparative analysis of the PKiKP
waveforms of Event 1 with other earthquakes in Central Asia, and present the P waves at the BVA
array from Event 1 in the frequency band of 1-4 Hz (For further details, please refer to comment 7).
Considering that the PKiKP waveforms of Event 1 are noticeably more complex than the
corresponding PcP waveforms and the source time function, and the similar complexity of the
PKiKP waveforms in the recordings from the KURA array for other earthquakes in the same region
was observed (Figure R2c). Therefore, we prefer that the PKiKP precursors observed in Event 1 are
more likely caused by anomalous structures at the ICB.

**Comment 2:**

While the modeling is impressively comprehensive, in the end I mostly take away the stated
conclusions "PKiKP distortions are mostly caused by the seismic anomalies near the ICB" and "the
distortion .. may be the combined effect of laminated and rough ICB" (and I'd add lateral variation
in the uppermost IC). It is very difficult to interpret layering rather than more disordered structure
from these observations as presented.

Reply: Thanks for your comment. We agree that it is very difficult to interpret layering rather than
more disordered structure from these observations as presented. The lateral variation of the structure
in the uppermost inner core could indeed lead to distortions in the PKiKP waveforms. We have
accordingly added the discussion in the revised manuscript in paragraph 15 and rephrased the

sentence in the conclusion to tone down this statement.

*“The distortion observed in the stacked PKiKP in Fig. 3c may result from the combined effect of*
*anomalies at ICB and lateral variation within the uppermost inner core.”*

**Comment 3:**

The authors could save the reader a lot of trouble by summarizing the observations at the start.
Essentially, they look at four earthquakes (3 + an explosion), comparing PKiKP to PcP - one in the
west shows a small 1s precursor to PKiKP on three tight arrays spread across a few hundred km,
several(?) events in the east either show a simple ICB or have PKiKP with a few extra cycles on the
end in the passband above 2 Hz. It literally took me hours, between the figures, text, and supplement,
to think I knew how the observations are being interpreted. It would be helpful, for example, to label
figures with which event and which array is shown in each frame.

*Reply: Thanks for your comments and suggestions. We have accordingly summarized the*
*observations at the start of paragraph 4. Indeed, Event 1 in the west shows significant double-peak*
*PKiKP waveforms on three small-aperture dense arrays (Figure 1b and Figure 3c), and three events*
*in the east either show a simple ICB (Figure 3b) or have PKiKP with a few extra cycles on the end*
*in the passband above 2 Hz (Figure 3c). We have accordingly added the sentence “Overall, Event 1*
*in the west shows significant double-peak PKiKP waveforms with a time separation of ~1.0 s on*
*three small-aperture dense arrays in Central Asia, while the other three events in the east either*
*feature a simple ICB or have PKiKP waveforms with a few extra cycles at the end in the frequencies*
*above 2 Hz.” in paragraph 7 in the revised manuscript.*

*Following your suggestion, we have accordingly labeled the updated Figure 3 with corresponding*
*the event and seismic array information in each frame.*

**Comment 4:**

The data surveyed remains unclear. It says “All 400+ events greater than M6.0 from 2012 to 2016”
(on what magnitude scales, in what area?) were examined (on all arrays?). But events 1 & 3 were
5.8, less than 6, and no area is defined. The text says all data screened with requirement SNR > 2,
next paragraph implies any visible arrival was retained, but for what data? Then it is noted “Found
two additional events” - why were these not in the original collection? I’m still left with the
impression the authors found 2 interesting events, then when pressed found two more, rather than a
more systematic survey, which is ok, but should be accurately described.

*Reply: We are sorry that the data survey part in previous versions is not very clear.*

*We have accordingly provided more details of seismic events in paragraph 3 from lines 66 to*
*73. Following your suggestions, we have specified the magnitude range and details of the events*
*distribution region. These earthquake magnitudes are range from MB5.7 to MB7.0 with a focal*
*depth of more than 80 km, which mostly distributed in the Hindukush, Myanmar, and the western*
*Pacific subduction zone regions. Considering the small aperture dense arrays in central Asia, which*
*have been in continuous operation since about 1994, then we collected data on over 440 events*
*occurring between January 1994 to December 2021 within the regions spanning 15-55 N° and 65-*

155 E°, which are recorded by those seismic arrays within epicentral distances of 50°. These
earthquakes are mostly distributed across the Hindukush, Myanmar, and the western Pacific
subduction zone regions.

Indeed, To select robust observations of PKiKP and PcP with high signal-to-noise ratios (SNR),
we retain waveforms either with an $\text{SNR} \geq 2.0$ for single traces or coherent signals on record
sections observed at dense arrays, and SNR is defined as the ratio of the peak amplitude of PcP or
PKiKP to the ambient noise amplitude of waveform window 5 to 10 s before the PcP or PKiKP
arrival. We found that most seismograms were excluded from further processing due to the high
noise levels significantly hinder the detection of PKiKP signals. For events with high SNRs in their
PcP and PKiKP seismograms, we additionally performed manual inspections to assess the quality
of valuable observations from small-aperture dense arrays. We found that despite some seismic
traces having an SNR below 2, PKiKP and PcP signals were coherent, thus still discernible.
Therefore, we retain those visually identifiable PcP and PKiKP signals recorded by small-aperture
arrays in Central Asia from these four events in Supplementary Table 1. The clarification of SNR
could be found in paragraph 4 from lines 79 to 90.

Thanks for pointing out the “...*find two additional events*...”, we are sorry for the confusion
caused by the unclear description, the two additional events are Events 3 and 4. To avoid
misunderstanding and improve the readability and clarity of the manuscript, we have accordingly
deleted the sentence “.....*we collect the data at the arrays from all the deep earthquakes in this*
*region and find additional two earthquakes*” in paragraph 7.

**Comment 5:**

I see reference to phase-weighted stacks and nth-root stacks. Both distort waveforms, precluding
estimation of the amplitude of weaker arrivals and thus comparison with synthetics. I'd greatly
prefer to see linear stacks, which would preserve waveform and frequency content.

Reply: Thanks for your comments and suggestions.

We agree that the linear stacks method could preserve waveform and frequency content,
achieving favorable effects in applications when the seismic array is relatively dense, and the signal-
to-noise ratio of the seismic phases is high. For instance, Tian & Wen (2017) adopted the linear
stacking to PKiKP and PcP observations recorded by the dense seismic array Hi-net, to constrain
the ICB structure beneath the Okhotsk Sea and western Pacific regions.

Indeed, the nonlinear stacking method, such as the nth-root process and phase-weighted stack
may distort signals (Rost & Thomas, 2002), thus we only use the nth-root stacking technique to
acquire the slowness of PKiKP and PcP phases in this study. The tf-PWS method is also a nonlinear
stacking method, which is widely used to reduce the noises and scattering waves and further enhance
the more coherent signals (Schimmel et al., 2011; Schimmel & Gallart, 2007). We conducted a
comparative analysis of the PKiKP waveforms obtained through linear stacking and tf-PWS
methods. From Figure R1, we found that the tf-PWS method indeed effectively enhance the SNR
of PKiKP, and the stacked PKiKP waveforms obtained from both methods were nearly identical.
Therefore, using the tf-PWS method to obtain stacked waveforms is also feasible in this study.

Figure R1. Comparisons of the stacked PKiKP waveforms in ICB anomalous region obtained
 through linear stacking and tf-PWS methods.

Schimmel, M., & Gallart, J. (2007). Frequency-dependent phase coherence for noise suppression in
 seismic array data. *Journal of Geophysical Research: Solid Earth*, 112(4), 1–14.
 <https://doi.org/10.1029/2006JB004680>

Schimmel, M., Stutzmann, E., & Gallart, J. (2011). Using instantaneous phase coherence for signal
 extraction from ambient noise data at a local to a global scale. *Geophysical Journal
 International*, 184(1), 494–506. <https://doi.org/10.1111/j.1365-246X.2010.04861.x>

**Comment 6:**

Figs 3b and 3c show very similar arrivals for all three earthquakes (I think Event 4 waveforms are
 only ever shown in Fig S13). These M6 events are nearly identical? The source time functions
 estimated in Fig S12 differs between events and in several cases is longer than the PKiKP and PcP
 arrivals. The arrival is somehow deconvolved? I strongly suspect that the events chosen here for
 sharp arrivals (and in Tian and Wen’s paper) are the small fraction that send a strong directivity
 pulse straight down to the CMB and ICB, and only a portion of the source time function is appearing.
 This case would not negate the conclusions, but might make the waveforms more sensitive to the
 take-off angle and azimuth, giving more possibilities to explain the observed differences and pose
 more difficulties interpreting the absolute and relative amplitudes.

Reply: Thanks for your comments and questions.

We are sorry for the confusion caused by the absence of corresponding events and seismic
 array information on each trace in the original Figures 3b and 3c. We have accordingly added labels
 in the updated Figures 3b and 3c, indicating the relevant events and seismic array information on
 each trace. Event 4 waveforms are shown in updated Figure 3b and updated Fig S7. Figure 3b shows
 the comparisons of stacked PKiKP and PcP waveforms in ICB normal regions (IN), the first trace
 is stacked PcP and PKiKP waveforms at the KURA array from Event 4, and the other seven traces

in Figure 3b are similar because they all the observations from Event 2. Figure 3c shows the
comparisons of stacked PKiKP and PcP waveforms in ICB anomalous region (IA), the first to third
traces are respectively stacks in KKA, KURA, and BVA seismic array from Event 3, and the fourth
and sixth traces are all stacks from Event 2, and the fifth trace is from Event 3. From Figures 3b and
3c, we could find that the stacked waveforms from all four moderate events (~ MB6) are different.

Indeed, the source time function (STF) of Events 3 is longer than their PKiKP and PcP
waveforms. And The PKiKP and PcP waveforms shown in Figures 3b and 3c have not been
deconvolved with the source time function, there are just the stacked results using the tf-PWS
method. Events 1-3 chosen here are dip-slip events with dip-angle around 45°, which is favorable
for sending a strong directivity pulse straight down to the CMB and ICB and exciting strong PKiKP
and PcP, and this type of source mechanism is indeed only a small fraction. It is possible that only
a portion of the STF of Event 3 is present in PcP and PKiKP phases.

We have added the statement *“And it is possible that only a portion of the STF of Event 3 is*
*present in PcP and PKiKP phases. That might make the PKiKP waveforms more sensitive to the*
*take-off angle and azimuth, giving more possibilities to explain the observed differences and posing*
*difficulties in interpreting the absolute and relative amplitudes.*

**Comment 7:**

Only event 1 has the 1s precursor, it's on all 3 western arrays. Is there no second event to check in
decades of recordings? PcP doesn't have to be clean to see if PKiKP often has the precursor. One
could also examine the direct mantle P waves to assess the presence or absence of a weak beginning
1s before a strong pulse in the source. For that matter, I'd like to see the mantle waves for Event 1,
which has the most anomalous observations, and the P wave should be presented in the same
passband, 1-4 Hz, as the ICB reflections.

Reply: Thanks for your comments and questions.

We have analyzed seismic recordings from the Hindukush region, spanning January 1994 to
December 2021, focusing on tens of moderate earthquakes with magnitudes ranging from 5.7 to 7.0
and depths greater than 80 km (Figure R2a). In our study, we indeed discovered PKiKP precursors
in the recordings from the KURA array for seven other earthquakes in the same region (Figures R2b
and 2c). The M5.9 earthquake in March 1998 was very similar to Event 1 in terms of location, depth,
and magnitude, so the PKiKP phase waveforms recorded by the BVA and KURK stations from the
two events, were strikingly similar (Figures R2b). But it is difficult to discern solely from this one
event whether the complexity of the PKiKP waveforms is caused by the shallow structures in the
source region or by an anomalous structure within the ICB. The other six earthquakes are situated
tens or nearly a hundred kilometers from Event 1, but they share striking similarities in their
sampling areas at the ICB with Event 1. This indicates that the complexity of the PKiKP waveforms
is likely not caused by the shallow structures on the source side, but rather by the anomalous
structure of the ICB. However, unfortunately, there are no observable PcP signals at those arrays in
Central Asia, due to contamination from high-frequency S-coda waves, thus we have not mentioned
these seven earthquakes in the manuscript. Considering that the PKiKP waveforms of Event 1 are
noticeably more complex than the corresponding PcP waveforms (Figure 1b, and Figure 3c) and the
source time function (supplementary Fig. 12a), and the similar complexity of the PKiKP waveforms

in the recordings from the KURA array for other earthquakes in the same region were observed
(Figure R2c). Therefore, we prefer that the PKiKP precursors observed in Event 1 are more likely
caused by anomalous structures at the ICB.

Thanks for your valuable suggestions. Indeed, the direct mantle P waves also could be as a
reference phase to access the source complexity in the study of ICB structures using PKiKP waves.
For instance, PKiKP/P amplitude ratios at a larger distance ($\geq 50^\circ$) were used to constraint the ICB
properties such as density contrast and velocity variation (Koper and Dombrovskaya, 2005), which
could mitigate the source complexity, shallow structural near source and station influence on PKiKP.
However, the epicentral distance of the PKiKP observations from Event 1 is less than 20 degrees in
this study, due to the influence of the mantle transition zone structure, the P-wave triplication will
appear at this epicentral distance (Figure R3a), and the time difference between the subsequent P-
waves and the first arrival is ~ 1 s. Following your suggestion, we present the P waves at the BVA
array from Event 1 in the frequency band of 1-4 Hz (Figure R3b), we found that the P waveforms
are so complex due to the mantle triplication at a distance less than about 30° , and it is difficult to
assess the presence or absence of a weak beginning 1s before a strong pulse in the source using the
P wave in this study.

Koper, K. D., and M. Dombrovskaya (2005), Seismic properties of the inner core boundary from
PKiKP/P amplitude ratios, *Earth Planet. Sci. Lett.*, 237(3–4), 680–694,
doi:10.1016/j.epsl.2005.07.013.

Figure R2. PKiKP observations recorded by seismic stations in Central Asia from earthquakes in
 Hindukush Region.

Figure R3. (a) Raypaths of P, PcP and PKiKP at different epicentral distances; (b) P observations
 recorded at BVA array in Central Asia from Event 1.

**Comment 8:**

Minor consideration:

Summing over the main array should average out lateral heterogeneity, producing a simpler apparent
arrival in the stack, for example in IN3 - do stacks of PKiKP on smaller subarrays have more
complex waveforms?

Reply: Thanks for your comment and question. We agree that summing over the observations on a
large aperture array could average out 3D lateral heterogeneity, and produce a simpler waveform in
the stack. Considering the China National Seismic Network are large- aperture dense array. We
divided it into seven sub-arrays with smaller apertures for the data stacking. Following your
suggestion, we have labeled the names of subarrays in the updated Figure 3. We found that the
stacked PKiKP waveform in the NCA is complex, while the PKiKP stacks in the other six sub-arrays
sampled the ICB in the IN3 region are comparatively simple. The stacked PKiKP waveforms
sampling the IN3 region might be more complex in smaller aperture arrays since PKiKP reflection
points within small-aperture arrays are very close to each other. Moreover, we also measured the
amplitude ratio of PKiKP/PcP on individual seismograms within seismic arrays and found that they
are comparable to the theoretical value in the IN 1-3 region (Supplementary Fig. 14b). This finding
also indicates that the ICB should be considered normal in IN 1-3 region.

REVIEWERS' COMMENTS

Reviewer #2 (Remarks to the Author):

Amazingly thorough response to the comments! My concerns and suggestions were well addressed.

My only comment at this point is the results are sufficiently significant for promotion to Nature Geoscience, if that remains an option, very interesting work!

NOTE: The reviewers' comments and suggestions are in black text, while our point-by-point responses can be found in blue text.

REVIEWER COMMENTS

Reviewer #2:

Amazingly thorough response to the comments! My concerns and suggestions were well addressed.

My only comment at this point is the results are sufficiently significant for promotion to Nature Geoscience, if that remains an option, very interesting work!

Reply: Thanks for your encouragement. We appreciate your thorough and helpful review. Your insightful and constructive comments have greatly improved the manuscript.